# Task-relevant representational spaces in human memory traces

Rebekka Heinen ⬡°*, Elias M. B. Rau°, Nora A. Herweg°, Nikolai Axmacher

Department of Neuropsychology, Institute of Cognitive Neuroscience, Faculty of Psychology, Ruhr University Bochum, Bochum, Germany

° These authors contributed equally to this work.
* rebekka.heinen@ruhr-uni-bochum.de

## Abstract

During encoding, stimuli are embedded into memory traces that allow for their later retrieval. However, we cannot remember every aspect of our experiences. Here, we show that memory traces consist of multidimensional representational spaces whose formats are flexibly strengthened or weakened during encoding and consolidation. In a series of behavioral experiments, participants compared pairs of natural images on either two conceptual or two perceptual dimensions, leading them to incorporate the images into representational 'spaces'. We found that representations from deep neural networks relate to both behavioral similarity and memory confidence judgements. Furthermore, we found that distances in task-relevant but not irrelevant spaces affected memory strengths. Interestingly, conceptual encoding did not impair subsequent rejection of similar lures, suggesting that task-irrelevant perceptual information remained in the memory trace. However, targeted memory reactivation following conceptual encoding deteriorated perceptual discrimination, indicating that it weakened the accessibility of perceptual formats. Our results demonstrate that representational formats are flexibly incorporated into memory, and more generally show how the organization of information in cognitive representational spaces shapes behavior.

## Author summary

Our brains create "memory spaces" that store information about our experiences. We found that these memory spaces are map-like and depend on task-relevant dimensions. These dimensions – i.e., different "representational formats" of a given stimulus – can be selectively strengthened or weakened during subsequent memory consolidation phases. In three behavioral experiments, healthy adult participants compared images based on conceptual (meaning) or perceptual (visual) features. Subsequently, they were presented old and new images and asked to indicate whether they had seen the image or not. Our results show

**Data availability statement:** All data needed to evaluate the conclusions in the paper are present in the paper and/or the Supplementary Materials. Data and analysis code to reproduce the main results presented in the manuscript are publicly available at https://osf.io/qn2gw/.

**Funding:** This work was supported by grants from the European Research Council (grant: 864164 to N.A.) and the German Mercator Research Center Ruhr (grant: Ex-2021-001 to N.A.). The funders had no role in study design, data collection and analysis, decision to publish, or preparation of the manuscript.

**Competing interests:** The authors have declared that no competing interests exist.

that the representations in deep neural networks can predict human similarity judgements, and that different models reflect different task-relevant representational formats. We also found that distances between image pairs affect memory strengths. Interestingly, focusing on conceptual details during encoding did not impair rejecting perceptually similar images later, suggesting task-irrelevant visual information was still retained. However, reactivating memories during sleep after conceptual encoding made it harder for participants to distinguish visually similar images. Our study shows how representational formats are incorporated into memory and indicate how cognitive maps shape our behavior.

## Introduction

When you recollect your previous birthday party, you may be able to mentally re-experience numerous details from this event – the taste of the cake, the melodies of the music, and the color of the wrapping paper – and these memories may already fuel imaginations of your next birthday. Episodic memories not only help us remember past experiences but also guide our future behavior. They rely on the reactivation of memory traces – i.e., internal representations of previously perceived events – which can be detected in the human brain via multivariate analysis methods such as representational similarity analysis (RSA) [1,2]. These memory traces guide behavioral memory decisions, yet not all information about an episode is stored. Remembering every detail would not only exceed the capacity of the brain but also contradict the need to generalize past information to future situations. However, it remains unclear which information from an episode is maintained in memory. According to some theoretical frameworks, memory traces are "sparse" and only contain novel or unexpected information, which is complemented at the time of retrieval by pre-existing semantic knowledge [3,4]. Alternatively, memory traces may initially contain rich sensory information that degrades slowly over time [5,6]. Finally, the amount and kind of information that is stored in memory traces may vary depending on the behavioral goals during an experience [7]. Here, we sought to test these competing hypotheses on the nature of the memory trace.

During perception, sensory information is hierarchically processed in the brain. For visual content, this hierarchy ranges from low-level perceptual information (e.g., edges and colors) to more complex features (e.g., textures and shapes) to semantic information. Here, we define perceptual and conceptual formats as representations of visual and semantic stimulus features, respectively, thus reflecting how a memory is stored rather than its content. Visual and conceptual features are extracted along the posterior-to-anterior extent of the ventral visual pathway (VVP) and lead to multiple coexisting representational formats of the same stimulus [5,8]. The processing hierarchy observed in convolutional deep neural networks (cDNNs) trained for object recognition provides a framework for understanding the representational formats [9–15]: previous studies have demonstrated that the representational structure of DNNs significantly corresponds to representations and processing steps along the

ventral visual pathway [15,16]. However, representations beyond the ventral visual pathway are not reflected in cDNN representations [17]. Thus, as representational formats gets more complex and semantic, there is need for other models to reflect human brain responses and behavior, for example corpus-based models of deep natural language processing (dNLP) that use text rather than images as input [18]. Indeed, recent studies found that representations of these language models matches human neural and behavioral similarities during language processing and narrative tasks [18–22]. Similarities between DNN and human representations are not limited to perception, but have been demonstrated similarity judgements [23,24], short-term memory [25,26], long-term memory [27] and perceptual and conceptual memory [28]. Thus, cDNNs and dNLPs provide different quantitative models of cognitive representations across various representational formats [8,27,29,30].

Representational similarities in individual formats correspond to distances in diverse representational "spaces", or cognitive maps, (with higher distances reflected by lower similarities) that can be employed for cognitive operations such as perception, memory, and reasoning [31,32]. While deep neural networks have been shown to model behavioral judgements relying on lower-level representational spaces [28,33] (e.g., defined by animacy or color), it is less clear whether these models can still reliably model representational formats in more complex and abstract spaces (e.g., the approachability of an animal). Since the relation between concepts and items in an abstract space is measured by the distance between the concepts, previous studies suggest the underlying maps to adhere to metric properties [31]. However, it remains to be elucidated whether task properties or interindividual differences in participant behavior create or influence these metrics. Thus, if participants rely on a metric cognitive space, similarities in abstract spaces should not only be present for metric task instructions but also in binary ratings. Information in cognitive spaces should also allow for generalization and inference to new concepts in the form of shortcuts. If we actually access abstract spaces depending on different representational formats, similarities in these spaces should be behaviorally relevant. In addition, idiosyncratic vs. shared (i.e., individual vs. collective) similarity structure in abstract cognitive maps remains to be explored.

While the neural basis and functional role of representational formats has been extensively studied during perception, fewer studies have investigated the formats of memory traces, i.e., of representations in the absence of sensory inputs [34,35]. Crucially, certain representational formats may be more relevant for subsequent tasks. These tasks may influence the availability and accessibility of information during later memory retrieval. Using deep neural network models (DNN) models to quantify perceptual and conceptual formats, it was recently shown that conceptual information is preferentially stored in a memory trace and supports later recall, while perceptual information is lost over time [25,27,28] (for review see [8]). This would suggest that memory traces are quickly transformed to only contain conceptual/semantic representational formats.

Prominent theoretical accounts have conceptualized these transformation processes as gist abstraction [36] or semantization [37,38]. These frameworks propose that transformation condenses detail-rich perceptual contents into more gist-like representations, because conceptual formats may be more generalizable and, therefore, more relevant for future behavior than perceptual formats. Processes of memory transformation and gist extraction have been linked to memory consolidation [39], which may start early after encoding [27,40] but is particularly pronounced during sleep [41]. Consolidation can be experimentally enhanced via targeted memory reactivation, i.e., by presenting auditory or olfactory cues during sleep that have been previously associated with stimuli [42,43]. However, few questions remain unanswered. Are perceptual features necessarily lost in favor of conceptual features, or do they remain silently available in the memory trace? In other words, is the representational format indeed subject to transformation (e.g., all sensory information is transformed into conceptual formats), or do multiple formats co-exist with some being gradually strengthened and others gradually weakened? And is this process influenced by task demands?

Indeed, various cognitive theories propose that representational formats depend on behavioral instructions during encoding. This has been suggested both in classical frameworks including levels of processing [44] and transfer-appropriate processing [45], but also in more recent theories based on reinforcement learning [46–48]. According to these

accounts, task demands during encoding may flexibly determine which representational formats are embedded into a memory trace. Recent work on cognitive maps demonstrates that people mentally "navigate" through abstract, conceptual spaces [49–53]. Task demands may influence which spaces are recruited and the path taken from one concept to the other may impact subsequent memory.

Further, task demands during retrieval may influence which specific representational format is required. In some cases, it may be necessary to reactivate perceptually detailed information from a memory trace to distinguish previously experienced stimuli from new items that are perceptually similar (pattern separation or differentiation). Other situations may require generalized information about previously experienced categories, independent of the perceptual details of a particular exemplar (pattern completion or generalization). These diverging retrieval demands would benefit from richer memory traces with multiple different representational formats that can be flexibly reactivated depending on situational needs [28].

Here, we conducted a series of behavioral studies to address several fundamental questions: How "shallow" or "deep" are representational formats of a memory – i.e., do they contain only a single representational format or multiple formats? How are these formats determined, how may they support different retrieval demands, and (how) do they change after encoding? We investigated the representational formats during encoding, consolidation, and retrieval, and tested (1) whether behavioral goals during encoding affect the representational "space" in which stimuli are embedded, and thus their subsequent memorability; (2) whether metric representational spaces are also employed when participants perform binary judgements during encoding, and whether these abstract spaces can be reconstructed using DNNs; (3) whether different representational formats serve different functional roles during retrieval; and (4) whether the formats that are incorporated into a memory during encoding can be subsequently modified via targeted memory reactivation.

## Results

### Selective embedding of memory traces into task-relevant representational spaces

In Studies I and II, participants were presented pairs of natural images of animals and were asked to judge the similarity of these images on either two perceptual dimensions (perceptual blocks), two conceptual dimensions (conceptual blocks), or one perceptual and one conceptual dimension (mixed blocks, only in Study I and removed from subsequent analyses, see Methods and Fig A in S1 Text; Fig 1B - 1C). During recognition, participants saw all images from the encoding phase together with an equal number of new items (Fig 1D) and indicated their confidence that a stimulus was old or new on a visual analogue scale (Fig 1D).

First, we confirmed that our task indeed led to different processing of perceptual and conceptual stimulus formats. To this end, we tested whether similarity judgements in the respective task-relevant perceptual and conceptual spaces corresponded to representational distances in specific DNN layers or models (Fig 1E). If participants employed perceptual spaces, rated distances should be reflected by distances in convolutional DNN layers that depend on sensory properties, while reliance on a conceptual format should be reflected by distances in a semantic model. Thus, we computed the similarity of each image pair in each network layer of a convolutional DNN (AlexNet [54]) and a dNLP (Google's universal sentence encoder [55]; USE). First, we investigated if convolutional cDNNs layers (conv), fully connected layers (fc) and the semantic model (USE) would reflect human similarity judgements according to rating task instructions (Fig 1F): We expected perceptual ratings to be related to visual cDNN and conceptual ratings to be related to semantic USE similarities. To this end, we analyzed layer-wise similarities using paired t-tests (one-sided), FDR corrected for multiple comparisons. Effect sizes are measured by Cohen's d. Indeed, representational distances in cDNN layers conv1 ($t_{101}=1.86$, $p=0.049$, $d=0.18$, perceptual: $0.024\pm0.1$ (M±SD), conceptual: $0.001\pm0.08$), conv3 ($t_{101}=2.58$, $p=0.017$, $d=0.25$, perceptual: $0.074\pm0.1$, conceptual: $0.038\pm0.09$), conv4 ($t_{101}=2.69$, $p=0.017$, $d=0.26$, perceptual: $0.07\pm0.09$, conceptual: $0.03\pm0.08$) and conv5 ($t_{101}=2.88$, $p=0.017$, $d=0.28$, perceptual: $0.08\pm09$, conceptual: $0.04\pm0.08$) predicted perceptual ratings better than conceptual ratings, while the reverse was true for fully-connected layer 8 ($t_{101}=-1.93$, $p=0.049$, $d=0.19$, perceptual: $0.10\pm0.09$, conceptual: $0.13\pm0.10$) and USE distances ($t_{101}=-2.15$, $p=0.037$, $d=0.21$, perceptual:$0.04\pm0.10$, conceptual: $0.07\pm0.09$).

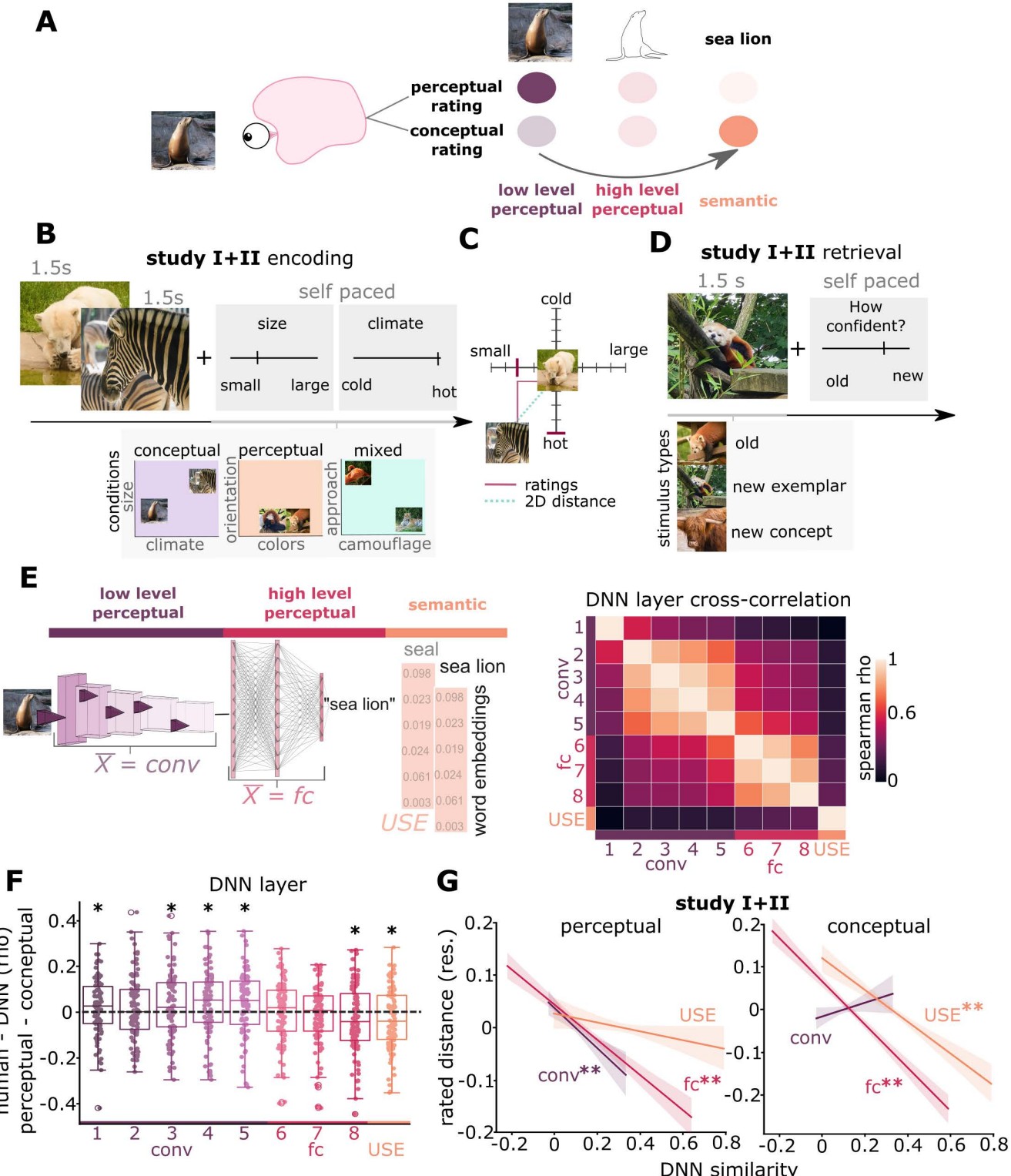

**Fig 1. DNNs reflect task-relevant representational formats.** (A) Theoretical framework: We hypothesize that memory traces consist of multiple representational formats whose accessibility can be flexibly adapted depending on task-demands. Perceptual tasks may increase the accessibility of perceptual features (left), while conceptual tasks may enhance processes of gist-abstraction and semantization (right). (B) Encoding phase in Studies

I+II: Participants encoded natural images of animals in either perceptual or conceptual conditions, emphasizing different task-relevant representational formats. In each trial, participants rated the similarity of image pairs on 2 different perceptual and/or conceptual dimensions via visual analogue scales. Study I comprised blocks with 2 perceptual features, blocks with 2 conceptual features, and mixed blocks. Study II contained only perceptual and conceptual blocks, during which 2 different sounds were played for later cueing during sleep (see Fig 4 for more details). (C) Representational distances were derived from similarity judgments in each single dimension and from the Euclidean distance of both judgments (2D distance). (D) Recognition phase in Studies I+II: Participants indicated their confidence regarding whether an image was old or new on a visual analogue scale. (E) Representational distances in low-level perceptual formats, higher-order perceptual formats and semantic formats were derived from a convolutional DNN and large language model, respectively. Layerwise and model cross-correlation reveals similarity clustering for convolutional layers, fully connected layers and distinct representations for the semantics model. (F) Layerwise correlation analysis reveals higher model-human rating similarity for early to mid cDNN layers (conv) to perceptual ratings (positive values) and higher similarity of late cDNN layers (fc) and the semantic model to conceptual ratings (negative values). (G) Using the averaged layer similarities, low-level perceptual and higher-order perceptual similarities predict rated distances during perceptual encoding. Reversely, conceptual ratings can be predicted by higher-order perceptual and semantic similarities. Linear model data are visualized after removing participant-wise estimated random effects. Dots represent participant averages. 95% confidence intervals, error bars indicate SEM.**, p<0.01, *, p<0.05.

Based on these results, the architectural differences between convolutional layers and fully-connected layers, and the cross-correlation matrix (Fig 1E), we then computed layer averages for the cDNN (see Methods for more details) of conv and fc layers. We then used the averaged conv, fc and the USE model similarities as predictors in a linear mixed model on rated distances of Studies I and II including a random participant intercept. Statistical significance was tested using a likelihood ratio test between a full model, including the fixed effect of interest, and a reduced model without the fixed effect of interest (all models included a participant intercept; see Methods for more details; beta weights of full linear models can be found in Table A in S1 Text). We found that distances in low-level perceptual space ($z = 4.54$, $\chi^2_{(1)} = 20.67$, $p < 0.001$; likelihood ratio test) and higher-order perceptual space ($z = 11.40$, $\chi^2_{(1)} = 129.51$, $p < 0.001$; likelihood ratio test) but not semantic space ($z = 1.87$, $\chi^2_{(1)} = 3.53$, $p = 0.06$; likelihood ratio test) predicted perceptual ratings (Fig 1G). Reversely, distances in higher-order perceptual space ($z = 15.80$, $\chi^2_{(1)} = 247.89$, $p < 0.001$; likelihood ratio test) and semantic space ($z = 7.30$, $\chi^2_{(1)} = 53.30$, $p < 0.001$) but not low-level perceptual space ($z = 1.76$, $\chi^2_{(1)} = 3.10$, $p = 0.07$; likelihood ratio test) predicted conceptual ratings (Fig 1G). To ensure that the range of rating responses used by participants did not influence the similarity between DNN-predicted and human behavioral judgements, we correlated participant-specific response ranges with the DNN matching, yielding non-significant relationships (Fig B in S1 Text). In addition, we tested how the DNN similarities reflected human ratings for each rating dimension separately (see Fig C in S1 Text for details and results). Taken together, these results suggest that human similarity judgements of perceptual or conceptual format features can be predicted using similarities from visual and semantic DNNs, respectively.

Next, we investigated whether these task-relevant spaces did not only reflect perceptual and conceptual formats during encoding, but whether they were also selectively incorporated into memory traces – i.e., whether the accessibility of individual items for recognition memory judgements was influenced by their task-relevant encoding task. First, we tested if participants were able to discriminate between old images, new exemplars and new concepts during recognition using one-sample t-tests and receiver-operator characteristics (ROC). We find confidence for all three stimulus types to be significantly different from zero (old: $t_{46} = 5.75$, $p < 0.001$ (SI), $t_{54} = 3.87$, $p = 0.003$ (SII); new exemplars: $t_{46} = 7.96$, $p < 0.001$ (SI), $t_{54} = 8.32$, $p < 0.001$ (SII); new concepts: $t_{46} = 14.93$, $p < 0.001$ (SI), $t_{54} = 16.84$, $p < 0.001$ (SII); Fig 2A). Participants successfully discriminated old images against lures (area under the curve (AUC) new exemplars: $t_{46} = 26.45$, $p < 0.001$ (SI); $t_{54} = 15.60$, $p < 0.001$ (SII); AUC new concepts: $t_{46} = 26.63$, $p < 0.001$ (SI); $t_{54} = 22.62$, $p < 0.001$ (SII)) and were better at rejecting new concepts as compared to new exemplars (AUC SI: $t_{46} = -15.28$, $p < 0.001$; SII: $t_{54} = -14.45$, $p < 0.001$; Fig 2B) with d-prime substantially above zero (old vs. new exemplars: d' = (mean±sd) 0.72±0.26; $t_{46} = 18.58$, $p < 0.001$ (SI), d'=0.63±0.28, $t_{54} = 16.34$, $p < 0.001$ (SII); old vs. new concept: $t_{46} = 20.03$, $p < 0.001$, d'=1.25±0.42, d'=1.23±0.46, $t_{54} = 19.07$, $p < 0.001$ (SII)). Next, we analyzed whether the encoding task of an item (perceptual or conceptual) affected the memory strength of that item. Using a linear mixed model of trial-wise memory strength with "encoding task" as a predictor and participant-specific intercepts as random effects, we did not find effects for more accessible memory traces following

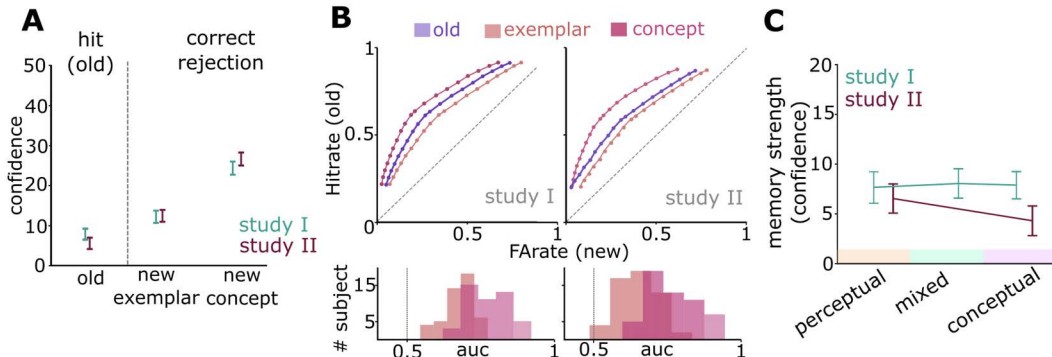

**Fig 2. Embedding of task-relevant representational formats into memory traces.** (A) Confidence ratings for old images and lures (new exemplars and new concepts). Higher values indicate higher confidence for hits (old items) or correct rejections (new items). (B) ROC and AUC for Study I and II indicate successful discrimination between old images and lures. (C) Memory strength did not depend on encoding tasks. Data are visualized after removing participant-wise estimated random effects. 95% confidence intervals. ***, p<0.001.

conceptual vs. perceptual judgements (Study I: z=-0.36, $\chi^2_{(1)}$ = 0.42, p=0.516; Study II: z=0.15, $\chi^2_{(1)}$ = 0.02, p=0.874; for Study II only uncued items were included in the analysis; likelihood ratio test; Fig 2C). This indicates that conceptual encoding instructions per se did not influence the subsequent accessibility of memory traces, pointing to a possible role of other factors such as the task-relevance of a particular format during encoding.

Next, we analyzed whether memory traces of individual items involved the fine-grained structure of task-relevant representational spaces, by testing whether representational distances in the respective task-relevant format during encoding influenced subsequent recognition memory performance. For each pair of images, we extracted both the task-relevant distances (i.e., the participant-specific ratings of their similarity) and task-irrelevant distances, which were calculated as the average distance of the same pair of images from a least squares solution (Fig 3A) over all rated images in all other spaces not rated by a given participant (Fig 3B). To test the validity of space reconstructions, we performed a premutation test, shuffling the rating response vectors while keeping the design matrices M constant to build a surrogate distribution of randomized relationships and ranked the empirical error of the LSQR model fit in the surrogate distribution (see Methods for more details; Fig 3C). Indeed, all reconstructed spaces reflect behavioral responses significantly better than chance (all p<0.001; Fig 3C). Interestingly, when comparing how global space maps (i.e., reconstructed based on the entire sample) reflect maps from single participants, we find higher correlations for conceptual as compared to perceptual spaces ($t_{101}$ = 17.40, p<0.001; Fig 3D), while the variance of participants global matching was comparable across spaces (all Levene tests p>0.05).

First, we ensured that rated distances were behaviorally relevant during encoding. Indeed, responses to images were given faster when their distances on relevant dimensions were larger (Study I: z=-11.28, $\chi^2_{(1)}$ = 126.69, p<0.001; Study II: z=-6.67, $\chi^2_{(1)}$ = 44.49, p<0.001; Fig 3E; see Fig D in S1 Text for more detail on the reaction times analyses). Second, we investigated how the rated distances influenced subsequent memory. To do so, we computed a linear mixed model to predict trial-wise memory strengths using the task-relevant distances (2D representational distances) as predictors and participant-specific intercepts as random effects. We found that distances in task-relevant but not task-irrelevant formats predicted subsequent recognition memory strength (Fig 3F). In both Study I+II, higher task-relevant representational distances predicted better recognition memory performance (Study I: z=4.16, $\chi^2_{(1)}$ = 17.32, p<0.001; Study II: z=5.74, $\chi^2_{(1)}$ = 33.02, p<0.001; likelihood ratio test). In Study II, this effect was significant for both the first (z=-4.65, $\chi^2_{(1)}$ = 21.65, p<0.001) and second image in a trial (z=-4.46, $\chi^2_{(1)}$ = 19.94, p<0.001); in Study I, it was significant for the second image (z=-4.27, $\chi^2_{(1)}$ = 18.27, p<0.001) and by trend for the first image (z=-1.64, $\chi^2_{(1)}$ = 2.71, p=0.099). Contrastingly, when including the average distance of the same images in task-irrelevant representational spaces as predictors in the linear

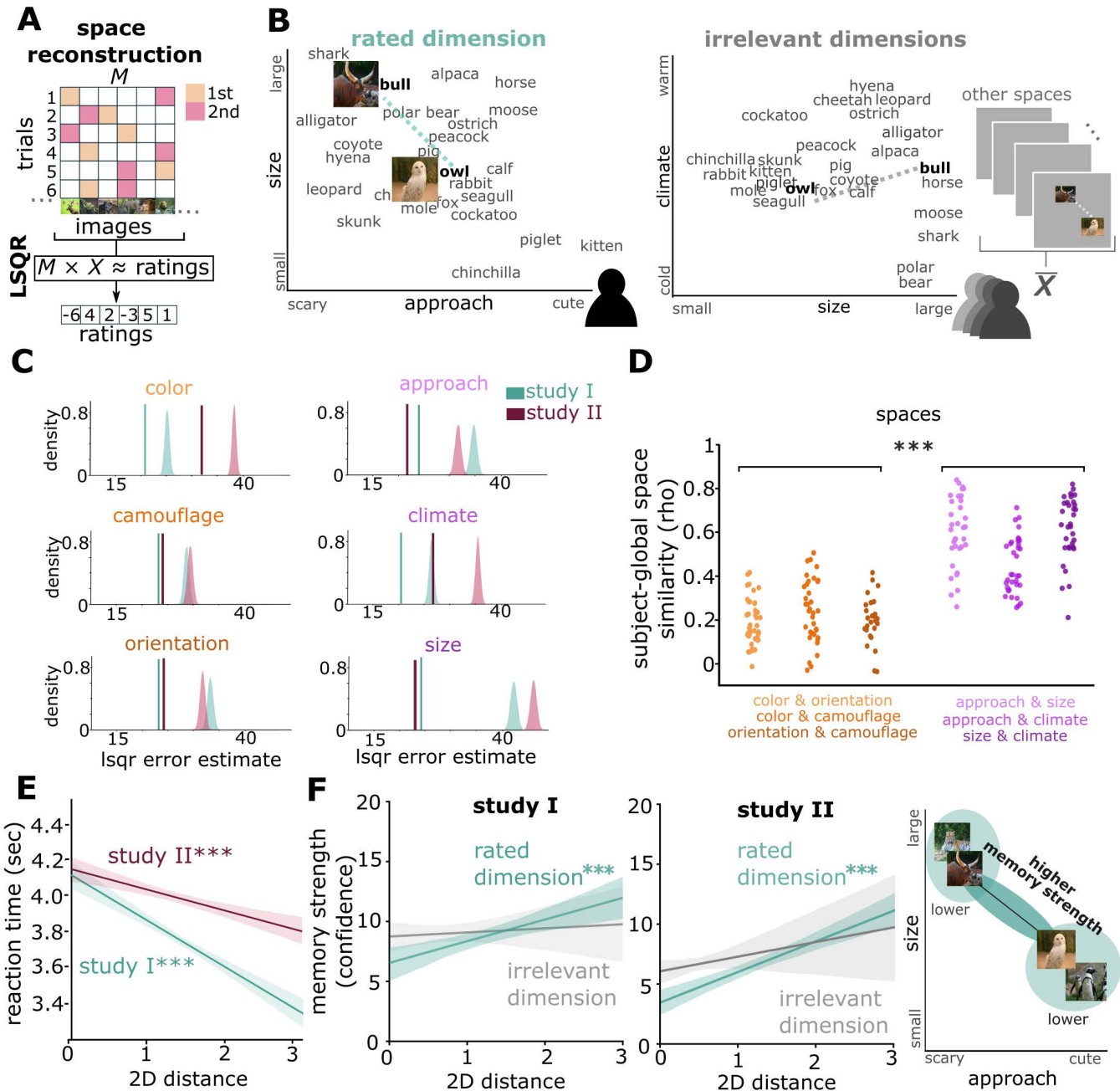

**Fig 3. Embedding of task-relevant representational spaces into memory traces.** (A) Cognitive space reconstruction. We used the ratings of each participant across all trials and all images to create a parametrization for each rated dimension. A sparse matrix (M) indicating which images were paired was used in a least squares solution (LSQR) given the participant ratings. Thereby each image receives a position value on the given dimension across all participant ratings. We can then combine two solutions (i.e., solution "approachability" and "size") to create a two-dimensional space. (B) Left: 2D distances in task-relevant encoding tasks were derived from participant-specific similarity judgments (figure shows space in example participant). Right: 2D distances in task-irrelevant spaces were computed as average distance in least square solutions of all other encoding tasks in all other participants (figure shows one corresponding task-irrelevant space across all other participants). (C) Validation of LSQR space reconstruction by comparing the estimated LSQR error to the error obtained from models that predict randomly shuffled behavioral ratings. (D) Subject to group level space correlations demonstrate how well the individual maps reflect the global space (a reconstructed space based on all ratings across participants). Perceptual spaces show more individual differences as compared to conceptual spaces which appear more general across participants. (E) Faster reaction times for higher rated distances during encoding. (F) Memory strength depends on 2D distances in task-relevant but not task-irrelevant encoding tasks with better memory for higher distances. Left: Study I, Middle: Study II, Right: schematic depiction. Data are visualized after removing participant-wise estimated random effects. 95% confidence intervals. ***, p<0.001.

mixed model, we did not find a significant effect (Study I: $z = -0.89$, $\chi^2_{(1)} = 0.80$, $p = 0.373$; Study II: $z = 1.25$, $\chi^2_{(1)} = 1.58$, $p = 0.208$; Fig 3F; see Fig E in S1 Text for separate results in the two encoding tasks). Thus, the irrelevant distances did not predict memory, independent of the processing format. While relevant distances predicted memory confidence for old images, we did not find an effect of relevant distance on correctly rejecting new exemplars as new (Study I: $z = -1.51$, $\chi^2_{(1)} = 2.31$, $p = 0.128$; Study II: $z = -0.78$, $\chi^2_{(1)} = 0.61$, $p = 0.433$).

These results show that task demands not only affect similarity judgements during encoding, but – more interestingly – that they also determine the representational space of an item's memory trace. Specifically, items that are encoded at higher task-relevant distances are remembered better, possibly reflecting effects of contextual deviance [56–59] and/or distinctiveness [60–62] (see Discussion).

## Employment of graded and metric representational spaces following binary ratings

In Studies I and II, the representational spaces of item-specific memory traces were defined by explicit distance ratings along graded similarity scales. This might have prompted participants to encode these images into cognitive spaces with metric distances. Next, we tested whether representational distances during encoding also affected memories if participants performed binary decisions in a forced-choice paradigm (Fig 4A) – indicating the use of cognitive spaces without explicit manipulation. During encoding in Study III, participants first saw a target image and then indicated which of two probe images was more similar to the target image. This decision was again based on either perceptual or conceptual features (Fig F in S1 Text). We observed high levels of accuracy in selecting the correct choice, with an even higher accuracy

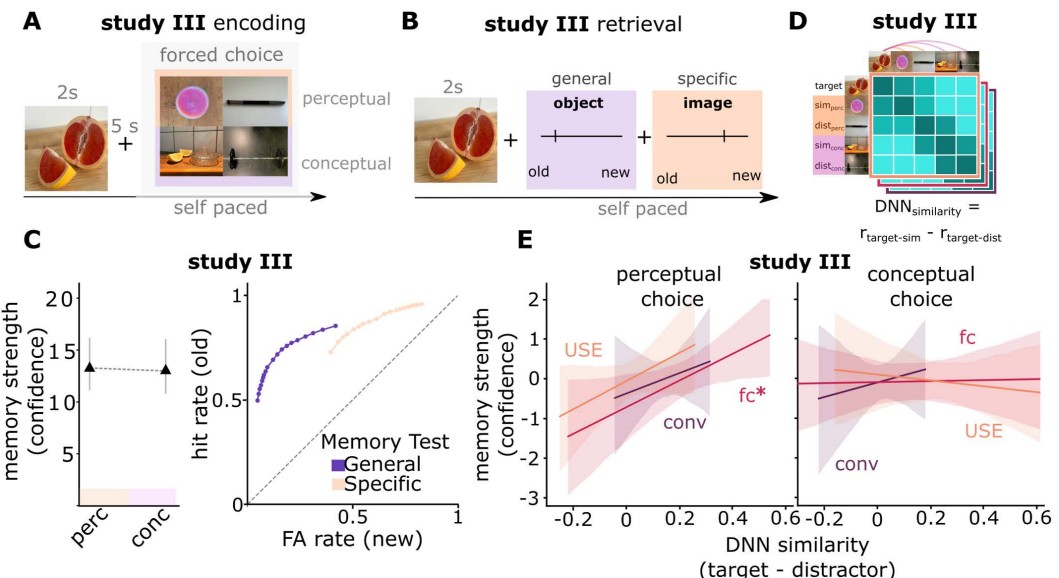

**Fig 4. Memory traces with DNN-based representational distances following binary similarity judgments.** (A) In Study III, participants performed forced-choice similarity ratings during encoding. Given a target image (blood orange), they chose a conceptually (lemon press) or perceptually (pink power pod) similar image from two options. (B) Participants first performed a general retrieval task (indicating whether a given image matched a previously presented category) and then a specific retrieval task (indicating whether the image matched the previously presented exemplar). (C) Encoding task does not influence memory strength. However, we find better discrimination performance in the general memory test. (D) Graded representational distances in low-level perceptual formats, higher-order perceptual formats, and a semantic format were derived from a convolutional DNN and the universal sentence encoder (USE), respectively. We computed the difference in DNN similarities between target image and similar (correct) choice vs. target and distractor (incorrect) choice, separately for each DNN level. (E) Task-relevant DNN-derived representational distances predict memory strength. Data are visualized after removing participant-wise estimated random effects. 95% confidence intervals, error bars indicate SEM. *, $p < 0.05$.

for conceptual vs. perceptual ratings (conceptual: M = 0.98, SD = 0.02; perceptual: M = 0.96, SD = 0.03; paired t-test across participants: $t_{27}$ = 4.23, p < 0.001; d = 0.82).

Next, we applied linear mixed models to test whether memory traces were embedded into task-relevant representational spaces, i.e., whether task-relevant distances during encoding predicted memory strength. In Study III, we refined the recognition task to separately address different recall demands either based on generalized or pattern-completed information ("Did you see an image from this category during encoding?"), or based on highly differentiated or pattern-separated information ("Did you see exactly this image during encoding?") (Fig 4B). Consistent with Studies I + II, we did not find an effect of encoding task on memory strength when using the average confidence of both the specific and general memory test for old images as predictors in a linear mixed model (z = -0.43, $\chi^2_{(1)}$ = 0.19, p = 0.66; likelihood ratio test; Fig 4C, left). In addition, we found better discrimination of old images against lures (AUC old vs. new exemplars: $t_{27}$ = 25.18, p < 0.001; AUC old vs. new concepts: $t_{27}$ = 24.03, p < 0.001) and better discrimination between lures (new exemplars vs. new concepts) for general memory (AUC $t_{27}$ = -9.52, p < 0.001) compared to specific memory (AUC $t_{27}$ = -29.01, p < 0.001; Fig 4C, right), while d-prime was substantially above zero (general memory: old vs. new exemplar: $d'$ = (mean±sd) 0.45 ± 0.51, $t_{27}$ = 4.75, p < 0.001; old vs. new concept: $d'$ = 1.87 ± 0.66, $t_{27}$ = 14.96, p < 0.001; specific memory: old vs. new exemplar $d'$ = 1.50 ± 0.47, $t_{27}$ = 16.88, p < 0.001; old vs. new concept: $d'$ = 2.21 ± 0.54, $t_{27}$ = 21.67, p < 0.001).

In order to identify the representational distances that guided participants' judgements in the absence of graded similarity ratings, we employed DNNs as in Study I and II to extract the similarities between pairs of objects in different representational formats. For each image pair, we thus computed the DNN similarities between the target stimulus and the correct choice image and between the target stimulus and the incorrect lure image (Fig 4D). The difference between these similarities was considered a proxy for distances in task-relevant representational spaces underlying the decision (see Fig F in S1 Text for more details).

We found that convolutional layers from the cDNN predicted response times during perceptual tasks, whereas fully-connected layers (cDNN) and the USE predicted response times during conceptual comparisons (Fig D in S1 Text). Importantly however, for perceptually encoded items, representational distances in higher-order perceptual space (z = 3.91, $\chi^2_{(1)}$ = 5.24, p = 0.022; likelihood ratio test), but not low-level perceptual (z = 2.91, $\chi^2_{(1)}$ = 0.55, p = 0.460; likelihood ratio test) or semantic spaces (z = 3.71, $\chi^2_{(1)}$ = 2.06, p = 0.150; likelihood ratio test; Fig 4E) predicted memory strength (general and specific memory tests combined; see Fig G in S1 Text for separate effects in the general and the specific memory test). For conceptually encoded items, representational distances did not predict subsequent memory strength (all p > 0.597).

Taken together, these results indicate that task-relevant representational distances can affect recognition memory performance, suggesting that memory traces of individual items are embedded in different representational spaces depending on encoding instructions. The linear relationship between task-relevant representational distances during encoding and memory strength (i.e., the accessibility of item-specific information for recognition memory judgements) suggests that memory traces involve coding schemes with metric cognitive maps in specific task-relevant formats (see Discussion).

## Effects of representational formats on different task demands during memory retrieval

We next explored the functional relevance of the different representational spaces for subsequent retrieval demands. Based on previous studies [63,64], we hypothesized that encoding in a conceptual space may be particularly beneficial for retrieval of generalized, perceptually invariant categorical information [65–67]. Conversely, encoding in perceptual spaces may be relevant when retrieval requires the rejection of similar but not identical exemplars (i.e., pattern separation).

We first analyzed whether the different retrieval instructions in Study III (general/specific) shifted memory decisions (Fig 5A). Indeed, in the general compared to the specific retrieval task, hit rates were significantly higher ($t_{27}$ = 11.55, p < 0.001; d = 1.50) and correct rejection rates of new concepts were significantly lower ($t_{27}$ = -12.35, p < 0.001; d = 2.59), suggesting a more liberal recognition memory threshold. Note that responses to new exemplars could not be directly compared because they required "old" ratings in the general task but "new" ratings in the specific task.

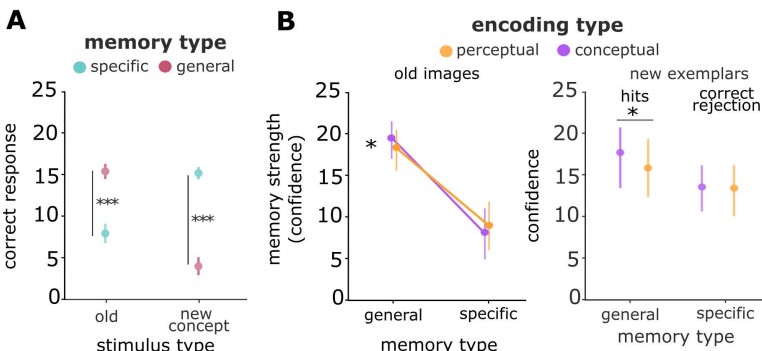

**Fig 5. Conceptual encoding improves generalized retrieval and does not impair specific retrieval.** (A) Different response criteria depending on retrieval tasks: General retrieval leads to higher hit rates but lower correct rejection rates of new concepts. (B) Left: conceptual encoding improves memory strength during general but not specific retrieval. Right: conceptual encoding facilitates correct recognition that new exemplars belong to previously presented categories during general retrieval and does not influence performance during specific retrieval. Data are visualized after removing participant-wise estimated random effects. Error bars indicate SEM. *, p<0.05, ***, p<0.001.

Next, we investigated whether performance in the general and specific memory tasks depended on task-relevant encoding tasks (Fig 5B). We used a linear mixed model to predict memory performance using retrieval task (general/specific) and encoding task (perceptual/conceptual) as predictors with a random participant intercept and including an interaction between retrieval task and encoding task. To test for significant differences, we used a likelihood ratio test between the full model containing the main effect and the interaction term, and the reduced model excluding the interaction. We found an interaction between retrieval task-demands and encoding task ($z = 2.69$, $\chi^2_{(2)} = 7.25$, $p = 0.007$; likelihood ratio test). Conceptual encoding improved memory in the general retrieval task ($z = 3.04$, $\chi^2_{(1)} = 9.27$, $p = 0.0023$; likelihood ratio test) but not the specific retrieval task ($z = 1.28$, $\chi^2_{(1)} = 1.66$, $p = 0.197$; likelihood ratio test; Fig 5B, left panel). Moreover, we found a significant interaction for new exemplars ($z = 2.53$, $\chi^2_{(2)} = 6.44$, $p = 0.011$; likelihood ratio test). Conceptual encoding benefited generalized memory decisions, i.e., hit rates towards new exemplars were higher than for perceptually encoded items ($z = 4.53$, $\chi^2_{(1)} = 20.52$, $p < 0.0001$; likelihood ratio test), while performance in the specific retrieval task did not depend on encoding instructions ($z = 0.47$, $\chi^2_{(1)} = 0.22$, $p = 0.6354$; likelihood ratio test; Fig 5B, right panel). Additional analyses indicated that employment of generalized information (in the general retrieval task) specifically benefited from lower representational distances in higher-order perceptual formats, while rejection of new concepts was impaired by lower distances in this format in both retrieval tasks (Fig H in S1 Text).

These results indicate that generalized information at test was more readily available for conceptually encoded images for which participants presumably relied more on gist-like representations. In other words, conceptual encoding created memory traces with generalizable representations that facilitated the correct recognition that new exemplars belong to previously presented categories. Thus, encoding stimuli into conceptual spaces improves both the accessibility of their memory traces (higher memory strength of previously presented exemplars) and their employment for generalized memory decisions (better recognition that new exemplars are members of a previously presented category). Interestingly, this does not occur at the expense of less perceptually distinct representations since performance in the specific memory test was not reduced, suggesting that conceptually encoded items are embedded into a "deep" memory trace with multiple different formats (even though task-irrelevant spaces do not seem to involve metric coding schemes; see Discussion).

## Selective strengthening of task-relevant formats during consolidation

So far, our results indicate that memory traces contain encoding tasks in distinct task-relevant representational formats. However, they also reveal an asymmetry, since conceptual encoding tasks benefited both general and specific retrieval,

whereas perceptual encoding tasks did not offer any particular advantage. Since memory consolidation has been suggested to strengthen conceptual representations [39], we explored its influence on the representational formats of memory traces. We employed targeted memory reactivation (TMR) in a between-subject design and tested whether cueing with a sound that was presented during either perceptual or conceptual encoding selectively enhanced the corresponding representational format (Fig 6A).

We first analyzed how TMR affected the memory strength of previously presented items. We compared conceptually encoded items in participants where these items were cued vs. conceptually encoded items in other participants where these items were not cued and vice versa for perceptually encoded items resulting in a between design. To this end we used a multi-level model with encoding task and cueing as predictors, and participant as random effect (Fig 6B). We observed a main effect of cueing ($z = -2.12$, $\chi^2_{(1)} = 4.53$, $p = 0.0333$; likelihood ratio test), indicating generally better memory for cued vs. uncued items, but also a significant interaction ($z = 4.38$, $\chi^2_{(2)} = 19.23$, $p < 0.001$; likelihood ratio test), indicating different cueing effects on perceptually vs. conceptually encoded items. Post-hoc tests revealed that cueing only improved the memory strength of conceptually encoded items ($z = -4.38$, $\chi^2_{(1)} = 19.23$, $p < 0.001$; likelihood ratio test) but not of perceptually encoded items, for which even a trend in the opposite direction was observed ($z = 1.88$, $\chi^2_{(1)} = 3.54$, $p = 0.06$; likelihood ratio test).

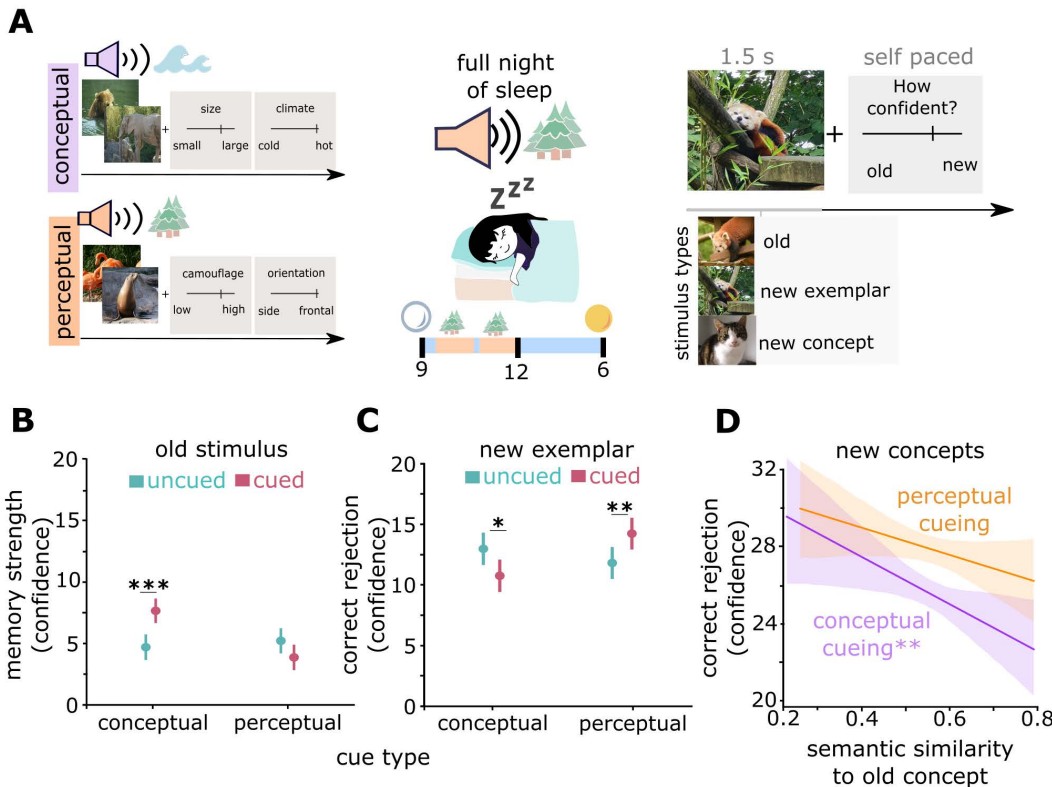

**Fig 6. Strengthening of task-relevant and weakening of task-irrelevant representational formats after TMR.** (A) TMR design: presentation of condition-specific auditory cues during conceptual and perceptual encoding blocks (e.g., coast sound and forest sound). One of these sounds was played again during the first half of the subsequent night, followed by a graded recognition memory test as in Study I. (B) Cueing of conceptual but not of perceptual formats increases memory strength. (C) Cueing of perceptual formats improves and cueing of conceptual formats impairs correct rejection of new exemplars. (D) Higher semantic similarities of new concepts to previously presented concepts impair correct rejection following conceptual but not perceptual cueing. Data are visualized after removing participant-wise estimated random effects. 95% confidence intervals, error bars indicate SEM. *, $p < 0.05$, **, $p < 0.01$, ***, $p < 0.001$.

Since our results thus far demonstrate that memory traces are selectively embedded in task-relevant representational spaces but also suggest that information in irrelevant formats remains present after encoding, further strengthening one of these formats may come at the expense of other formats. Even though conceptual encoding by itself does not impair specific retrieval of perceptual information (Fig 6B), one may hypothesize that further strengthening of conceptual formats during consolidation reduces the perceptual specificity of item representations, increasing false alarms to novel exemplars (i.e., lures) of previously encoded categories. We thus implemented a linear mixed model for memory ratings of similar lures with encoding condition and cueing as predictors (Fig 6C). This model revealed a significant interaction ($z = -3.32$, $\chi^2_{(2)} = 13.11$, $p = 0.0003$; likelihood ratio test), showing different effects of cueing on correct rejection of similar lures for conceptually vs. perceptually encoded items. Follow-up tests showed that cueing of conceptually encoded items deteriorated the identification of similar lures ($z = 2.38$, $\chi^2_{(1)} = 5.70$, $p = 0.017$; likelihood ratio test), while cueing of perceptually encoded items improved similar lure identification ($z = -2.77$, $\chi^2_{(1)} = 7.66$, $p = 0.0056$; likelihood ratio test). Consistent with our results in Study III, conceptual encoding by itself (i.e., without cueing) did not deteriorate lure detection ($z = -0.35$, $\chi^2_{(1)} = 0.12$, $p = 0.726$; likelihood ratio test). These results indicate that cueing can selectively increase either conceptual representational formats (improving recognition of old items but deteriorating rejection of lures) or perceptual formats (with no effect on later memory but more accurate rejection of lures).

Next, we sought to investigate whether cueing of conceptually encoded items also impairs the ability to identify novel concepts, if they are similar to previously encoded categories. We used the semantic similarities, quantified by the dNLP, of novel concepts to all previously encoded concepts as a predictor in a linear mixed model (Fig 6D). The results revealed that cueing conceptually encoded items impaired identification of semantically related novel concepts ($z = -2.65$, $\chi^2_{(1)} = 7.01$, $p = 0.008$; likelihood ratio test). This effect was not found after perceptual cueing ($z = -1.60$, $\chi^2_{(1)} = 2.56$, $p = 0.11$; likelihood ratio test). As a control, we confirmed that the influence of low-level perceptual and higher-order perceptual similarities on the identification of novel concepts did not differ between cueing conditions (Fig I in S1 Text).

These results indicate that representational formats of memory traces are not fully determined by the initial task-relevance during encoding, but that they can be further and selectively strengthened during subsequent consolidation stages. Our findings further show that depending on the task demands to either recognize previously presented items or to reject similar lures, these consolidation processes may exert either beneficial or detrimental effects.

## Discussion

Generative memory theories [4,68,69] propose that memories are inherently constructive. They suggest that memory traces are rapidly transformed from perceptual into conceptual/semantic representational formats, and that missing information is supplemented from prior knowledge or schemas at the time of retrieval [3,70]. Other frameworks propose that memory traces contain richer representations in multiple formats whose relative accessibility can be flexibly determined by various factors during encoding, consolidation, and retrieval [5,71]. The aim of our study was to systematically investigate the role of representational formats in memory traces: Are transformations of a memory trace indeed unidirectional, such that perceptual formats are irreversibly lost in favor of semantic information, or do multiple formats coexist? Our study yielded three main results; first, we showed that memory traces consist of distinct representational formats that are determined by task demands during encoding and affect the accessibility of information during subsequent retrieval. The linear relationship between representational distinctiveness and memory strength suggests that stimuli are embedded into metric encoding spaces that are determined by objective item features which correspond to different formats of DNNs. Second, we found asymmetric functional roles of these different encoding spaces for subsequent retrieval demands. While conceptual encoding benefits retrieval during generalized task conditions, it does not impair performance during specific retrieval, suggesting that perceptual information is still maintained in a "deeper", richer memory trace. Finally, we observed that consolidation selectively strengthens task-dependent representational formats and weakens task-irrelevant formats, leading to "shallower" memory traces.

First, our findings demonstrate the heuristic potential of DNNs to investigate the representational spaces underlying similarity judgements and the selective embedding of these spaces into memory traces. Previous studies have shown that DNNs can be used as a proxy for perceptual and semantic formats during memory tasks [25,27,28,72,73] and behavioral similarity judgements [73,74], and our results indicate that the same models can account for similarity judgements during encoding and their impact on subsequent memory. Importantly, we found that distances from cDNNs and from a natural language model contributed differently. While distances based on perceptual cDNN features predicted perceptual ratings, reaction times, and memory strengths, distances of deeper cDNN layers and of a semantic model predicted behavioral outcomes related to conceptual formats. These results emphasize the usefulness of DNN models as proxies of different representational formats – even in behavioral outcomes – and contribute to the idea of multiple formats in a memory trace.

The levels of processing (LOP) theory posits that memory performance depends on the depth of processing during encoding, yielding better memory for conceptually (deep) as compared to perceptually (shallow) encoded stimuli. Contrary to this prediction, neither our study nor other investigations have found a consistent main effect of encoding format on later memory While unexpected, our results are in line with other studies finding no general effect of processing depth on subsequent memory [75,76]. Several factors may explain this discrepancy. First, while many LOP studies predominantly use words as stimuli, images are less frequently employed [77], and the extent to how task instructions engage differential processing depth of natural images remains unclear. Additionally, in our paradigm, image durations were relatively long (1.5 seconds) as compared to presentation durations in the order of only a few hundred milliseconds [78] and we can therefore not rule out that conceptual features were equally extracted during perceptual encoding conditions. Indeed, during object recognition, semantic category can be decoded after a few hundred milliseconds in inferior temporal cortex [79] suggesting that automated extraction of conceptual features were possible in both encoding conditions. Interestingly, a recent study showed that image features, measured as reconstruction error of a cDNN, and not the orientation to specific features predicts later memory [80]. Third, our paradigm involved pairwise comparisons of images, introducing a working memory component not typically present in classical LOP tasks, which rather present single items or multiple items simultaneously [81]. Interestingly, research has shown that both perceptual and conceptual formats are retained during working memory maintenance periods after initial encoding [25], suggesting that both types of representational formats contribute to neural representations of individual images.

Importantly, the relational judgement of image pairs afforded participants to process and respond to stimulus relationships (i.e., rated distance), rather than focusing on individual item or artificial visual features (i.e., a circle around a concept for perceptual judgements instead of features of the concept itself) [82]. Accordingly, encoding depth may be less concerned with the individual stimulus features as in classical LOP studies, but more likely depended on the relational structure in task-dependent representational spaces. Further, pairwise comparisons likely necessitate the extraction of high-level meaning to evaluate pairwise item relationships. While it could be shown that the attribution of meaning influences subsequent memory [83], this may contribute equally to judgements based on perceptual or conceptual representational formats.

We observed robust evidence for a role of task-relevant representational spaces. Specifically, the distances of task-relevant but not irrelevant dimensions predicted the memory strength of both perceptually and conceptually encoded items. Thus, task-demands [7], but not the mere depth of processing, shapes which features are prone to decay, and which are preserved and even strengthened to support later memory. Interestingly, we found better memory if stimuli were encountered at larger representational distances in their relevant encoding space. How can we explain the higher accessibility of stimuli encoded at higher task-relevant distances? First, one may assume that strengthened memory for more dissimilar items could be caused by higher prediction errors [84–86]: While the current reference position, e.g., of a shark in the "approachability" space, might lead to the expectation of further dangerous animals, the presentation of a chinchilla would lead to greater prediction errors. However, this explanation cannot account for our finding that not only the second, but also the first image in pairs with larger distances was remembered better. For the same reasons, contextual deviance

(as in the von Restorff effect [59]) is less likely the main factor driving this result [87,88]. Alternatively, higher dissimilarities may promote processes of pattern separation [89], which may further facilitate the disambiguation of similar images during retrieval. However, we did not find that higher task-relevant encoding distances facilitated the correct rejection of similar lures. Previous studies showed that the hippocampus encodes representational distances in task-relevant cognitive maps [50,90]. Here, we extend these findings by showing that these distances are actually relevant for subsequent memory, and thus contribute to theories bridging spatial navigation and episodic memory [31,91]. Specifically, we found that representational distances were linearly related to the accessibility of items for later memory, even if items were not originally encoded using graded distance ratings (Study III). This suggests that participants employed cognitive maps with interval-scaled (metric) rather than rank-based (ordinal) coding schemes [92], in line with the original concept of cognitive maps to promote inference during physical navigation [93].

While our study suggests that information represented in abstract spaces may be metric, it is possible that other measures of representational geometry are at play. For instance, graph-like representational structures [90] might be used in our task, although ultimately these may also employ a metric (i.e., two steps on the graph are equivalent to a certain distance). The use of a visual scale might have enforced the use of a metric space in our task; however, we found similar results for Study III where participants indicated similarity via binary responses. It would also be interesting to see if the spaces in our study enable shortcuts by predicting relations for leave-out concepts. Even more interesting is the overall fit of the individual spaces to the global group space, indicating an overlap between interindividual similarity judgements and a collective representational structure in the employed abstract maps. Specifically we find that perceptual spaces were more individualized than conceptual spaces, which is supported by a recent finding suggesting that perceptual ratings are highly subjective and individually different [94]. The more conceptual or abstract these spaces become, the more similar they might be across people. This raises exciting new questions about the nature of interindividual differences in representational spaces – will conceptual spaces become more similar to a global group space during development, as individuals learn and interact with these concepts? Which factors (i.e., prior knowledge, attention, reward, etc. influence how similar your own space is to the global map?

Notably, while all three studies presented here provide converging evidence for a selective strengthening of task-relevant but not task-irrelevant representational formats, we also consistently found that conceptual encoding by itself did not come at the expense of reduced identification of similar lures. This suggests that the benefit of conceptual encoding may not (only) lie in deeper encoding and faster integration with pre-existing semantic knowledge, but (also) in the formation of "deeper" and more versatile memory traces that can be flexibly employed for both specific and generalized retrieval demands. Even though the accessibility of perceptual information in conceptually encoded memory traces does not depend on (in this case, task-irrelevant) perceptual distances, and is thus not organized in a "metric" cognitive map (see above), it remains available even following a night of sleep (Study II). Future studies will need to investigate whether more extended consolidation processes across longer time periods promote the formation of "shallower" traces in purely conceptual/semantic formats, leading to a more pronounced decline in the availability of perceptual information.

Interestingly, we observed detrimental effects of conceptual as compared to perceptual encoding on lure identification following contextual TMR. This effect has been linked to complex encoding paradigms, which demonstrated both advantageous but also detrimental effects on memory [95]. This further supports current findings on contextual reinstatement effects during sleep [96], especially for complex, multidimensional memories [97]. Furthermore, our results suggest that enhanced consolidation processes induce a "pruning" of memory traces that removes information in task-relevant representational formats. Previous studies have demonstrated that cueing during sleep is associated with reactivation of the associated memories [98–102]. After cueing conceptually encoded items, participants relied more on gist-like representations, causing a decreased performance in the rejection of both visually similar lures and semantically similar concepts. These findings are in line with results that reactivation during sleep facilitates generalization processes of features that are shared across several encoded contents [103–105], which in turn increases false alarm rates for similar lures [106].

However, while this may be interpreted as evidence for a unidirectional transformation of memory traces from perceptual into conceptual formats during consolidation, we also found that cueing of perceptually encoded items improved the correct rejection of visually similar lures. This suggests that TMR – and possibly, memory consolidation and sleep-related reactivation more generally – selectively enhances task-relevant [42,95,107] and reduces task-irrelevant representational formats, rather than generally transforming perceptual into conceptual formats. Recent work indeed suggests reactivation-induced forgetting during sleep: While target memories were enhanced, this enhancement was accompanied by forgetting of irrelevant competing memories [108,109]. Thus, while multiple representational formats coexist in memory traces after encoding, these formats can be selectively strengthened and weakened by consolidation processes [5].

Our results are solely based on behavioral measures. While this does not allow us to directly demonstrate representational formats of memory traces on a neural level, we understand behavior as a measurable expression of the underlying memory trace [1]. Additionally, our study emphasizes the utility of behavioral paradigms during (arguably) ecologically valid tasks for inferring complex properties of human cognitive representations [110]. Nevertheless, we propose that follow-up studies should aim to investigate these representations more directly using neuroimaging and/or electrophysiological recordings. Specifically, previous studies have emphasized a role of the hippocampus for learning [111] and for representing multidimensional abstract spaces [31,50,112]. Thus, it would be interesting to investigate whether hippocampal circuits can be flexibly adapted to represent the respective task-relevant formats due to their high degrees of mixed selectivity [113], or whether different formats are represented in different hippocampal regions (e.g., along its long-axis) [114–116]. In addition, testing our sleep effects within the same participant would offer insights into interindividual differences in consolidation during sleep. Moreover, while previous studies investigated representations in newly formed abstract spaces [49,117,118], participants in our study performed similarity and memory judgements in pre-existing spaces of semantic and perceptual relationships, which may involve interactions of hippocampal distance representations with distinct regions along the VVP. Finally, neuroimaging studies are ideally suited to investigate "dormant" memory traces during states without direct behavioral readouts, such as awake/resting periods, sleep, or while concurrent cognitive tasks are performed. Thus, these studies can explore the mechanisms underlying the "pruning" of memory traces during consolidation. Future studies should also focus on testing additional DNNs and their relation to human similarity judgements since model architecture (such as recurrent connections between layers [12]), and even starting weights can influence DNN representations [119].

Taken together, we propose that both perceptual and conceptual representational formats coexist with various factors such as attention, prior knowledge, and task-demands determining the accessibility of a specific format thereby shaping the apparent transformation of memories over time and contexts.

## Materials and methods

### Ethics statement

Participants gave written informed consent and received study credits as compensation. All three studies were approved by the institutional review board of the Faculty of Psychology, Ruhr University Bochum (Study I: #730, Study II: #554, Study III: #682), and were conducted in accordance with the declaration of Helsinki.

### Sample

We tested healthy adult students from Ruhr University Bochum (Germany). In Study I, we tested 47 participants (age M = 22.9, SD = 4.0, 38/9 female/male). In Study II, the sample comprised 55 participants (age M = 23.05, SD = 3.57, 44/11 female/male) after excluding one participant who woke up during the night and recognized the sound played. In Study III, we tested 28 participants (age M = 23.17, SD = 3.22, 16/12 female/male) after excluding three participants due to technical issues. Analysis of DNN contributions to prediction of encoding distances was done on the combined data from Studies I and II (n = 102).

## Behavioral tasks

Data for all three studies were collected online. We implemented the tasks for Studies I and II using jsPsych [120] and Cognition (https://www.cognition.run/). In Study III, we implemented the task using Psychopy3 [121] and Pavlovia (https://pavlovia.org/).

We chose images from the THINGS database [122], a large database of natural images from different categories (e.g., animals, objects, and food), specifically built to make DNN training sets more naturalistic and applicable to neuroscience. Items in this database represent an ecologically valid reflection of human-relevant everyday objects. For Studies I and II, we used animal images only to avoid effects of animate/inanimate differences during later similarity analyses. We excluded images containing multiple animals or containing humans. Since the encoding phase of Study III consisted of predefined forced choice trials, the conceptual and perceptual triplets (target and two choice options) showed not only animals, but equal amounts of objects from other categories contained in the THINGS database, namely food, tool, container, vehicle, and a compilation of individual objects from miscellaneous categories (e.g., musical instruments, electronic devices, and toys). For each participant, the amount of target objects from each category was kept equal within encoding blocks and across perceptual and conceptual conditions. During recognition, new concepts were selected from the same pool of categories and in equal amounts. Note that for copyright reasons figures include proxy photographs taken by the authors and are not part of the THINGS database.

Studies I and II consisted of two parts, an incidental encoding task (Fig 1B) and a recognition task on the following day (Fig 1C). During encoding, participants rated conceptual or perceptual similarities of a series of image pairs drawn from different categories. In each trial, we consecutively presented two images for 1.5 seconds each, followed by two self-paced ratings which required either conceptual or perceptual similarity judgements. Participants rated similarities of the last presented image in relation to the first image on a visual rating scale (-50–50; Ratings of zero indicating the images score the same on the rated dimension, i.e., polar bear and walrus both live in the same cold climate). Deciding on the rating dimensions, we chose judgements that would either require an exemplar (i.e., image; Does an animal look straight at you or to the side for orientation) to actually judge its similarity for perceptual ratings, while for conceptual ratings, we chose judgements that would only require the animal name (i.e., elephant and mouse for real world-size). Conceptual judgements included size ("Is the animal smaller or larger in the real-world?"), climate ("Does the animal live in a colder or a hotter climate?"), and approach ("Is the animal more approachable or less?"). Perceptual judgements included orientation ("Is the animal more facing towards the viewer or more away from the viewer?"), colors ("Does the image contain warmer or colder colors?") and camouflage ("Is the animal concealed more or less?"). Since we wanted to examine if perceptual or conceptual formats, or the combination of both, would strengthen subsequent memory, we divided the six rating dimensions into three different block types. Each type consisted of two dimensions from either the same or different conditions (perceptual, conceptual, mixed). We presented 336 images of 84 animals in total, with each animal being shown in one of the three block types only (e.g., the four different images of giraffes were all shown in perceptual blocks). Block types of each animal and image pairing were randomly shuffled between participants.

In Study II, we investigated the role of consolidation in the transformation of representational formats. We used the same design as in Study I but paired conceptual and perceptual ratings with condition-specific sounds (Fig 5A). During each encoding block, we continuously played one of two distinct nature context audios [123]. One audio was composed of ocean waves and sounds of the sea. The other audio consisted of forest sounds, like the brushing of leaves in the wind, frogs croaking, and crickets chirping, excluding bird sounds to avoid interference with our animal classes. One of the two sounds was played again during post-encoding sleep to reactivate memories associated with the sound [124,125]. Each audio file was randomly assigned to be played either during conceptual or perceptual encoding blocks. We decided to exclude mixed rating blocks, as there would be no clear conclusion regarding how consolidation of both rating dimensions should impact subsequent memory performance, while simultaneously, this would have increased the number of trials to cue during sleep. In the night following the encoding session, participants were asked to play a custom-made audio file

using their smartphone, starting the file when they would start trying to fall asleep for the night. In standardized studies applying targeted memory reactivation, EEG data would be monitored online to specifically play sounds during slow wave sleep [42,126]. Due to practical reasons and to ensure contact-free participation in our study during the COVID-19 pandemic, we decided to play an audio file timed in accordance with the expected higher amount of slow wave sleep during the first half of the night [127–129]. Audio files were composed to play either the perceptual or conceptual sound during the first half of sleep with long periods of slow wave sleep, which are highly relevant for consolidation [126,128]. The audio file started with a masking white noise to habituate participants to the volume, in order to avoid waking them up due to the sudden onset of the actual cue playing during sleep. Sixty minutes after audio onset, the cue was played for 30 minutes, followed by 60 minutes of masking noise. The cue was then played again for 30 minutes. If participants had been awake for longer periods or had to use the bathroom until 2 a.m., they were instructed to restart the audio, though no participant reported waking up prior to this time. The next morning, participants were asked whether they heard any specific sounds during sleep, woke up, or stayed awake for longer time periods. If participants recognized the sounds being played during sleep or if they did not have a full night of sleep between encoding and recognition, we excluded these participants from later analysis (n = 1). We also excluded trials for Studies I and II with reaction times greater than 150 seconds.

The day following encoding, participants performed a recognition memory task which was identical for Studies I + II (Fig 1C). We presented the same images again ("old"), mixed with a total of 336 new images. Novel images were either unknown images of previously presented animal species ("new exemplars") or images of animal species that had not been presented before ("new concepts"). Each image was presented for 1.5 seconds. Afterwards, participants indicated their confidence of having seen the image before on a continuous rating scale (-50–50) from "sure old" (-50) to "sure new" (50). All 336 old images, 168 new exemplars, and 168 new concepts were shown in a random order. For analysis, we rescaled the confidence ratings to a range of 0–50 so that higher values would reflect higher confidence for both the old images and the lures.

To examine the role of binary rather than gradual similarity ratings and of different retrieval demands, we implemented different encoding and retrieval tasks in Study III (Fig 3A - 3B). Instead of subjectively rating the similarity of image pairs during encoding, participants performed forced-choice similarity judgements of triplets (three serially presented images). Each triplet began with a 2s presentation of the target image, followed by a 5s maintenance period during which a fixation cross remained on screen. After maintenance, two horizontally arranged images were presented on either side of the fixation cross (left and right choice option). Participants were instructed to select the image that was more similar to the initially presented target on either perceptual or conceptual dimensions. Similar to Studies I and II, participants performed blocks of the encoding task in which either perceptual (image) or conceptual (object) features were relevant for performing the task.

In order to ensure reliable and valid responses across participants, we carefully constructed trials for each target image where either a perceptually or conceptually similar option was shown together with an unrelated distractor. Stimuli were selected based on similarity predictions from the cDNN. For each of the 120 target images, we manually selected images to form a conceptual and perceptual triplet by identifying images from the THINGS database which predominantly shared feature similarities with the target on either early (convolutional) DNN layers (perceptual, e.g., similar colors, shapes: blood orange - pink powder) or later (fully-connected) DNN layers (conceptual, e.g., similar object category: blood orange - lemon press). Accordingly, for each triplet, we selected distractor images that did not share the respective features with the target image. To ensure a meaningful difference in similarity of target and choice options, we quantified the difficulty of each triplet by computing the difference in the similarity between the target and the matching stimulus ($r_{sim}$) and the similarity between the target and the distractor ($r_{dist}$). For the distribution of scores on each visual and semantic DNN layer, see Fig F in S1 Text. We restricted our analysis to triplets with >75% accuracy across participants, which led to exclusion of n = 5 perceptual triplets.

After successfully performing the forced choice task, participants received monetary reward feedback of two cents (low) and 10 cents (high). Reward conditions did not elicit behavioral differences across conditions in neither encoding or

recognition, nor their congruency in encoding-recognition, and are thus not further discussed. For all analyses of Study III encoding, N = 3,161 trials were considered after excluding trials with incorrect forced choice response (n = 86) and trials with reaction times exceeding four standard deviations above the participants' average (n = 44).

Similar to Studies I + II, participants engaged in a recognition memory test on the following day (Fig 3B). During this test, 120 old images, 120 new exemplars, and 120 new concepts were presented for two seconds each. In each trial, participants consecutively indicated their confidence of having seen the object (general retrieval) and having seen the specific image (specific retrieval) using two separate scales. Participants indicated general memory before specific memory, similar to the procedure in [28]. Note that while for old images (old concept, old image) and new concepts (new concept, new image), correct responses in the two retrieval tests were on identical sides of the slider scale, this differed for new exemplars, which required "old" responses in the general retrieval task but "new" responses in the specific retrieval task.

As for the encoding task, we excluded trials with reaction times exceeding four standard deviations above the participants' average reaction time (n = 120 perceptual trials, n = 120 conceptual trials), resulting in a total trial number of N = 9,494 recognition trials across all participants.

### Lure discrimination performance

First, we tested whether participants confidence (response on the visual analogue scale) for correct responses to old stimuli and lures was significantly different from zero using one-sample t-tests. To test if participants successfully discriminated old images from lures, we computed receiver operator characteristics (ROC). To this end, we divided our continuous confidence rating scale into bins of 5 and tested whether the area under the curve (auc) across all confidence bins was above chance (0.5) using one-sample t-tests (Fig 2B; Fig 4C). In addition, we computed d' tested if d' was different from zero using one-sample t-tests.

### Linear mixed models

We analyzed all data from the three studies using linear mixed models of the statsmodels package [130] in Python 3.8. To determine the influence of different factors on behavioral performance, we set up a full model containing all fixed effects, and participant as a random effect. We then used a likelihood test to determine whether the exclusion of the factor of interest significantly reduced the explained variance compared with the full model. For main effects with one predictor, the reduced model included a constant as fixed effect (1). For main effects with two or more predictors, the reduced model contained all other main effects except the effect of interest. For interaction effects, the full model contained all main and interaction effects. However, main effects from interaction models were not interpreted. For each fixed effect, we report z-statistics from the initial estimation of the model and the respective likelihood derived from model comparison ($\chi$2) and its significance level (p). Beta weights resulting from the full models are shown in Table A in S1 Text (for the main results) and Table B in S1 Text (for analyses from the Supplement). For post-hoc estimation of condition differences, we ran separate models on data split by the factor of interest, and then compared the likelihood of the reduced model with only a random intercept. For visualization purposes of individual fixed effects, we plotted the residuals of mixed models missing the fixed effect of interest. Finally, individual models were used to determine the behavioral performance in conceptual vs. perceptual encoding tasks as well as memory performance of old images, new exemplars, and new concepts during recognition.

### Constructing task-relevant versus task-irrelevant distances

To compare the effect of relevant and irrelevant distances in Studies I + II, we first constructed individual one-dimensional spaces for each dimension by computing a least squares solution on a sparse matrix of all image ratings on the respective dimension over all participants (Fig 3A). For this, we first created a design matrix M with dimensions of the number of ratings by the number of images containing zeros. Next, M was filled for each rating with 1 or -1 indicating if an image

was first (1) or second (-1) in the rating trail. The design matrix M was then converted into a compressed sparse column matrix. A least-squares solution was then used to solve the linear system of equations on our design matrix M given the ratings done by participants. Two least squares solutions can then be used to compute the distance of each image pair in the respective 2D space. We then use these spaces to extract the distances of image pairs on dimensions that the participant did not rate the image pair on, thus creating an "irrelevant" distance as an average over all spaces. The irrelevant distance in this case thus corresponds to a distance that was not relevant for the participant during the task. In contrast to this, we define the relevant distance as the similarity judgment of each image pair that the participant performed during the task. For example, when a participant rated the similarity of a panda and a giraffe on the dimensions of climate and size (rated distance), we used the average distance from the least square solutions of all other spaces (climate + approachability; size + approachability; color + camouflage; orientation + camouflage; color + orientation) as the irrelevant distance.

To test if our reconstructed spaces indeed predict behavioral responses better than chance, we computed a random distribution of image x rating pairings based on 10.000 repeats and extracted the corresponding LSQR error estimate. We then plot the estimated error from the real ratings and the distribution of the shuffled random data (Fig 3C)

Next, we investigated how each individual participant space would relate to the global space based on all participant ratings. We computed the participant to global space similarity using spearman rho correlations of the reconstructed global space similarity matrix and the reconstructed individual similarity matrix of each participant (Fig 3D). We then tested if this participant to global space fits differed between spaces and format using a t test (perceptual vs. conceptual). Next, we tested if the variance of participant – global fits would differ between spaces by computing Levene tests for the participant – global correlations between the six spaces.

## DNN similarities

We used a pre-trained convolutional deep neural network, "AlexNet" [54], as implemented in the Caffe framework [131] in Python 2.7, and abstract semantic representations from the Google Universal Sentence Encoder [55] using tensorflow [132] in Python 3.8. (Fig 1E). We chose AlexNet for mainly two reasons: First, previous studies have linked the representations of this network to human neural and behavioral similarities [15,25–27], second while other studies make use of deeper cDNNs such as VGG-16 [133], AlexNet consists of only 8 layers, which offers deeper insights into the "blackbox" of DNNs and higher levels of interpretability even of intermediate layers.

To analyze cDNN similarities, we first generated features for each layer of the AlexNet for all images. In each layer we averaged across the spatial dimension, retaining just one value per feature. We then correlated the features of each image with all other images using Spearman's *rho* correlation, resulting in eight image-by-image correlation matrices, one for each layer. These correlation maps represent the similarity of each image to all other images across the layers of the cDNN, starting with similarities based on early visual information in convolutional layers and concluding with higher-order visual information of fully connected layers (Fig 1E).

We averaged the correlation maps of convolutional and fully connected layers to simplify later interpretation of differences concerning behavioral performance and reducing the number of statistical tests to correct for. In addition, a cross-correlation of layers and models suggested high similarities between convolutional layers and between fully connected layers, while also demonstrating that the semantic model represents a metric distinct from the visual cDNN (Fig 1E). Thus, we obtained two correlation matrices from AlexNet, which served as our markers of low-level perceptual representations (convolutional layers) and higher-order perceptual representations (fully connected layers), while the USE model contains abstract semantic feature formats. We computed word embedding vectors for all concept labels retrieved from the THINGS database and correlated each concept embedding vector with all others to generate a concept-by-concept correlation matrix.

Pairwise image similarities from perceptual and semantic DNNs were used to quantify the degree of overlap between pairwise images, given specific features (Fig 1F). For Studies I + II, we capitalized on the behavioral ratings. Participants

were instructed to rate the relation of both images on two dimensions, where the second image was rated relative to the first image. We expected the effect to be dependent on the rated dimension (conceptual vs. perceptual) with higher contributions of early (convolution) cDNN layers to perceptual ratings, and of late (fully-connected) cDNN layers and USE to conceptual ratings. This analysis was done on the layer-wise correlation maps to investigate the effect in more detail across the DNN gradient. For each layer, we performed a one-sided paired t-test (scipy package [134]) on fisher-z transformed average correlations between participant ratings of perceptual and of conceptual dimensions (conceptual, perceptual, and mixed for Study I; conceptual and perceptual for Study II) and the similarities of the respective cDNN layer and USE, resulting in nine paired t-tests. We then corrected for multiple comparisons using FDR-correction. We computed the difference between perceptual and conceptual similarities in each layer to plot layer-wise relations to similarity judgements. To validate the choice of similarity metric used (Spearman's correlations) we repeated this analysis using Pearson correlations and obtained similar results (see Fig C in S1 Text). Next, we used layer-averaged, and fisher-Z transformed similarity matrices as predictors in mixed models for rated distance during encoding, and for memory strength during recognition.

In addition to combining perceptual and conceptual rating dimensions in the DNN analysis, we performed the same analyses (layerwise correlations and linear mixed models using DNN similarity averages to predict human ratings) for each of the six dimensions separately (Fig C in S1 Text).In addition, we tested how the semantic model would relate to the human similarity judgements of each of the single conceptual dimensions and tested for differences between the three spaces (size, climate, approach) using t-tests, corrected for multiple comparison (Fig C in S1 Text). Finally, we aimed to exclude that the range of participant responses would influence the relation of similarity judgements to the DNN similarities by computing the correlation between DNN – participant rating correlations and the participant ranges (maximum – minimum response) using Pearson correlation (Fig B in S1 Text).

To determine the degree of format-specific processing during memory formation, we constructed trial-specific measures indicating the distinctiveness of similarity judgements of task-relevant feature formats. Each image was considered a point in 2D space, of which we computed its trial-specific Euclidean distance. The first image was the point of reference with coordinates [0,0], while the coordinates of the second image were the given ratings on each scale [rating1, rating2]. For Studies I and II, we computed the similarity of a new exemplar or new concept to all images previously shown during encoding on conv and fc layers for perceptual similarity and the similarity of a new concept to all previous concepts and extracted the highest similarity for later analyses. For Study III, we computed the similarity of a new exemplar to its matched images during encoding. For new concepts, we extracted the highest similarity of a new concept to all old concepts from encoding.

In Study III, we extracted our measure of distinctiveness from the pairwise similarities across DNN-derived layer predictions for all three images of a triple (target, similar choice, distractor choice) (see Fig F in S1 Text). We took the similarity of the target image with the similar choice ($r_{sim}$) and the distractor choice ($r_{dist}$) and used their difference ($d = r_{sim} - r_{dist}$) as quantification for triplet (dis-) similarity. Both measures of trial-distinctiveness quantify the degree of difficulty during a given trial (e.g., small distances of animals indicate similar features), making the magnitude and direction of the given rating more difficult. Likewise, low triplet-distinctiveness indicates that both choice options (similar and distractor) are approximately equally similar to the target image, making the forced choice task more difficult. This was done for all images of a triplet in Study III and all pairs of old images and their corresponding new exemplars. This was also done to estimate the distance of new concepts to the bulk of all previously seen concepts by fetching the highest similarity to all old concepts (i.e., smallest distance).

## Supporting information

**S1 Text. Supplementary Information contains additional analyses and supplementary figures A–I and supplementary tables A–B.**
(PDF)

## Author contributions

**Conceptualization:** Rebekka Heinen, Elias M. B. Rau, Nora A. Herweg, Nikolai Axmacher.

**Data curation:** Rebekka Heinen, Elias M. B. Rau.

**Formal analysis:** Rebekka Heinen, Elias M. B. Rau.

**Funding acquisition:** Nikolai Axmacher.

**Investigation:** Rebekka Heinen, Elias M. B. Rau, Nora A. Herweg.

**Methodology:** Rebekka Heinen, Elias M. B. Rau, Nora A. Herweg.

**Project administration:** Rebekka Heinen, Elias M. B. Rau, Nora A. Herweg, Nikolai Axmacher.

**Resources:** Rebekka Heinen, Elias M. B. Rau.

**Software:** Rebekka Heinen, Elias M. B. Rau, Nora A. Herweg.

**Supervision:** Nora A. Herweg, Nikolai Axmacher.

**Validation:** Rebekka Heinen, Elias M. B. Rau.

**Visualization:** Rebekka Heinen, Elias M. B. Rau.

**Writing – original draft:** Rebekka Heinen, Elias M. B. Rau, Nora A. Herweg, Nikolai Axmacher.

**Writing – review & editing:** Rebekka Heinen, Elias M. B. Rau, Nikolai Axmacher.

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
