## [Decision Letter · Decision Letter 0]

6 May 2025

Task-relevant representational spaces in human memory traces

PLOS Computational Biology

Dear Dr. Heinen,

Thank you for submitting your manuscript to PLOS Computational Biology. After careful consideration, we feel that it has merit but does not fully meet PLOS Computational Biology's publication criteria as it currently stands. Therefore, we invite you to submit a revised version of the manuscript that addresses the points raised during the review process.

Please submit your revised manuscript within 60 days Jul 06 2025 11:59PM. If you will need more time than this to complete your revisions, please reply to this message or contact the journal office at ploscompbiol@plos.org. Please include the following items when submitting your revised manuscript:

We look forward to receiving your revised manuscript.

Kind regards,

Paul Bays

Academic Editor

PLOS Computational Biology

Lyle Graham

Section Editor

PLOS Computational Biology

**Additional Editor Comments :**

As you will see, the Reviewers have positive things to say about your manuscript but also raised some concerns that will need to be addressed before a final decision can be made.

**Journal Requirements:**

At this stage, the following Authors/Authors require contributions: Rebekka Heinen, Elias MB Rau, Nora A Herweg, and Nikolai Axmacher. Please ensure that the full contributions of each author are acknowledged in the "Add/Edit/Remove Authors" section of our submission form.

5) We notice that your supplementary Figures, and Tables are included in the manuscript file. Please remove them and upload them with the file type 'Supporting Information'. Please ensure that each Supporting Information file has a legend listed in the manuscript after the references list.

Potential Copyright Issues:

i) Please confirm (a) that you are the photographer of 1, 2, 3, 5A, S1A, S4A, S5A, and S7, or (b) provide written permission from the photographer to publish the photo(s) under our CC BY 4.0 license.

ii) Figures 1A, 2B, and 5A. Please confirm whether you drew the images / clip-art within the figure panels by hand. If you did not draw the images, please provide (a) a link to the source of the images or icons and their license / terms of use; or (b) written permission from the copyright holder to publish the images or icons under our CC BY 4.0 license. Alternatively, you may replace the images with open source alternatives. See these open source resources you may use to replace images / clip-art:

7) Please amend your detailed Financial Disclosure statement. This is published with the article. It must therefore be completed in full sentences and contain the exact wording you wish to be published.

1) State what role the funders took in the study. If the funders had no role in your study, please state: "The funders had no role in study design, data collection and analysis, decision to publish, or preparation of the manuscript.".

**Reviewers' comments:**

Reviewer's Responses to Questions

Reviewer #1: Heinen et al. use a behavioral and modelling approach to show that memories are encoded into representational spaces whose formats are dependent on task demands. Notably, representations derived from DNNs correlate with behavioral judgments where distances within task-relevant representational spaces influence memory. This is an impressive and well-executed study that offers valuable insights into the nature of human memory representations using an combination of behavioral experiments and deep neural network modeling. The analyses are thorough, the experimental design is clever, and the manuscript is clearly written, making a strong contribution to our understanding of computational cognitive neuroscience. I mainly have a few clarification questions and suggestions below.

1. The Nature of Representational Space - metric?: Do these spaces strictly adhere to metric properties? Do the authors think there is a real metric space we're all using, or is it more individualized, and doesn't matter exactly what number it is? The typical use of rank correlations in RSA suggests this it might not matter exactly what the metric is.

2. Surprisingly good human-model link given the task: Conceptual space in humans in this study is distance based on size and climate (2D). In the DNN space it is distance in general (high dimensional). How is it that only two conceptual features is surprisingly similar to the DNN's general distance presumably with many semantic features? I understand it probably captures part of the variance, but do you think you're capturing most of it, or just a part of it?

3. Participant Variability: It would be interesting to know a bit more about the range of participant responses and how this might relate to the correlations observed with the DNNs. For instance, is the consistency of the correlation with DNNs robust even if individual response ranges vary? What were the range of responses? Did people use the whole range? Does that matter to the correlations with the DNNs?

4. Visualizing Human-Model Correlations: The human-model correlation plots - why not show the individual dots? Are there too many? It would be nice to be able to see the distribution of the correlations, though not necessarily in all plots.

5. Grouping DNN Layers: the convolutional and fully connected layers were grouped - could the authors justify this (rather than looking at all layers) in methods or somewhere please.

6. LSQR: As a minor point, it might be interesting to briefly discuss potential ways to validate the LSQR sparse matrix regression, perhaps by predicting out of sample stimulus distances.

Overall, this is a high-quality manuscript. Addressing these minor points would likely enhance the clarity and impact of this work.

Reviewer #2: The present manuscript, "Task-relevant representational spaces in human memory traces," reports on a study investigating how task demands during encoding, consolidation, and retrieval shape the structure and format of memory traces. Across a series of behavioral experiments, participants compared pairs of natural images (primarily animals) along either conceptual (e.g., size, climate) or perceptual (e.g., color, orientation) dimensions. These comparisons embedded the images into multidimensional "representational spaces," the structure of which was quantitatively modeled using deep neural networks (DNNs) to approximate both perceptual and semantic features.

A key finding is that distances between items in task-relevant representational spaces—those dimensions emphasized during encoding—predicted subsequent memory strength: items that were more distinct along these relevant dimensions were better remembered. In contrast, distances in task-irrelevant spaces showed no such predictive effect. Interestingly, conceptual encoding did not impair the ability to reject perceptually similar lures, suggesting that perceptual details remain accessible within the memory trace even when not emphasized during encoding.

The study also examined the role of consolidation through targeted memory reactivation (TMR) during sleep. Sounds previously paired with either conceptual or perceptual encoding were replayed during sleep, selectively enhancing memory for the corresponding representational format. However, strengthening conceptual representations through TMR appeared to reduce access to perceptual detail, as evidenced by impaired discrimination of similar lures.

Taken together, the results suggest that memory traces are flexible and encompass multiple coexisting representational formats, with their relative accessibility shaped by both encoding instructions and consolidation processes. This flexibility enables memory to support both specific (detail-rich) and generalized retrieval demands.

The manuscript addresses an important and timely topic. However, the methods section is currently difficult to follow due to the number of experimental variations and manipulations. A more structured and transparent presentation is recommended. In addition, some assumptions and analytic decisions require further justification. Below, I outline specific major and minor comments and suggestions for improvement.

Major Comments

• Implementation of conceptual vs. perceptual dimensions:

It is not fully clear in how far size judgements are less perceptual‘ than orientations of the animals. Given that the conceptual vs. perceptual manipulation is critical for the presented studies, the authors should clearly justify their choices for this manipulation.

• Use of DNNs as Ground Truth for Representational Formats:

In this study, DNNs are used as models to differentiate perceptual from conceptual processing. However, the introduction does not provide sufficient justification for (a) why DNNs should serve as a proxy for human processing, and (b) why the specific network implementations and layers chosen are appropriate models for distinguishing between perceptual and conceptual representations in human. Without this, the statement “Taken together, these results demonstrate that our rating task manipulated participants’ employment of task-relevant spaces” appears overstated. A more cautious interpretation would be that human similarity ratings show correlational associations with different DNN layers, in a manner consistent with theoretical assumptions about perceptual and conceptual processing hierarchies in DNNs. Here and throughout the manuscript the authors should opt for more descriptive interpretations in the results section and use more interpretative speculations only in the discussion section.

• Recognition Memory Analyses:

Recognition memory outcomes should be reported in more detail. Confidence ratings should be presented separately for old and new items—and, if applicable, for different categories of new items (e.g., similar lures vs. novel foils). Furthermore, it would be valuable to determine whether recognition performance was above chance. One approach would be to dichotomize the confidence scale (e.g., using the midpoint as a neutral threshold) and classify responses into hits, misses, correct rejections, and false alarms. This would allow evaluation of recognition performance using standard signal detection theory metrics (cf. Macmillan & Creelman, 2004).

• Use of the Term “Encoding Space”:

“We first analyzed whether the encoding space of an item (perceptual or conceptual) affected the memory strength of that item.“

The phrase “encoding space” implies an interpretive conclusion that may not be justified at the level of the manipulated variables. In the experiments, participants performed one of two encoding tasks (conceptual vs. perceptual). Whether these tasks give rise to separable “encoding spaces” is an inference based on the results, not a direct manipulation. This distinction should be reflected in the manuscript by using more neutral language during result reporting and reserving theoretical terms for the interpretation sections.

• Lack of Memory Advantage for Conceptual Encoding:

The finding that recognition memory did not differ between perceptual and conceptual encoding is surprising, given the well-established levels-of-processing framework (Craik & Lockhart, 1972), which posits that deeper (conceptual/semantic) processing enhances memory retention. Numerous studies have since demonstrated superior memory for semantically encoded material. The manuscript should address this discrepancy and provide possible explanations for the absence of a conceptual encoding advantage.

• Clarification of TMR Design and Figure 5B:

I may have missed a critical detail in the description of the TMR manipulation—if so, I apologize for the misunderstanding. However, I find the logic behind Fig. 5B somewhat confusing. As I understand it, each encoding block (conceptual vs. perceptual) was paired with a distinct sound, and during sleep, only one of these sounds was replayed. If, for instance, the sound associated with conceptual encoding was played during sleep, then only conceptually encoded items could have been cued. Consequently, the uncued items in that condition would necessarily be perceptually encoded. Under this setup, the left panel of Fig. 5B effectively compares conceptual vs. perceptual items—given that cueing and encoding conditions are fully confounded—making it difficult to attribute observed effects to cueing rather than to the original encoding condition.

This issue becomes even more complex when comparing Fig. 5B with Fig. 1G. Fig. 1G presents memory performance by encoding condition and suggests that conceptual encoding results in lower memory strength (with values below 5). In contrast, Fig. 5B shows memory scores for cued conceptual items around 8. Moreover, based on Fig. 1G, perceptual encoding appears to lead to better memory performance than conceptual encoding. These discrepancies are difficult to reconcile and raise questions about the internal consistency of the findings.

In my view, a clearer and more transparent reporting of the memory results—especially to facilitate comparisons across studies within the manuscript—is essential.

Minor Comments

• Figure 4B:

The y-axis should begin at zero to improve interpretability and avoid misleading visual effects.

• Confidence Scale:

The confidence rating scale is insufficiently described in the methods and figures. Please provide the numeric range and explain how it maps onto qualitative confidence judgments (e.g., “sure new” to “sure old”).

• Consistent Axis Scaling Across Figures:

Figures presenting the same metric (e.g., memory strength, confidence ratings) should use consistent axis ranges across the manuscript to facilitate direct visual comparison.

• Inclusion of Response Time Results:

The reported finding that similarity judgments were faster when the distance between items along relevant dimensions was larger (Supplementary Fig. 2A) is important and should be included in the main text.

**Have the authors made all data and (if applicable) computational code underlying the findings in their manuscript fully available?**

Reviewer #1: Yes

Reviewer #2: Yes

PLOS authors have the option to publish the peer review history of their article (what does this mean? ). If published, this will include your full peer review and any attached files.

**Do you want your identity to be public for this peer review?** For information about this choice, including consent withdrawal, please see our Privacy Policy .

Reviewer #1: No

Reviewer #2: No

**Figure resubmission:**

**Reproducibility:**



---

## [Decision Letter · Decision Letter 1]

1 Sep 2025

Dear Dr. Heinen,

We are pleased to inform you that your manuscript 'Task-relevant representational spaces in human memory traces' has been provisionally accepted for publication in PLOS Computational Biology. 

Note that Reviewer 1 has a suggestion for a minor edit to a figure legend which you could make before submitting your final version.

Best regards,

Paul Bays

Academic Editor

PLOS Computational Biology

Lyle Graham

Section Editor

PLOS Computational Biology

Reviewer #1:

Reviewer #2:

Reviewer's Responses to Questions

**Comments to the Authors:**

Reviewer #1: Thanks to the authors for providing a comprehensive response that goes beyond addressing all my questions. Congratulations for a strong paper and I recommend that it should be accepted.

My only extremely minor comment is in Fig 3D the legend states "Subject to group level space correlations demonstrate how well the individual maps reflect the global space." and it is not immediately clear what the global space means. My initial thought was it's an "objective" space, but it is actually the space based on all participants. Maybe a minor edit like: "…how well the individual maps reflect the global space (a reconstructed space based on all ratings across participants).

Reviewer #2: The authors were very responsive to the questions and feedback in my previous review and replied adequately. I have no further comments.

**Have the authors made all data and (if applicable) computational code underlying the findings in their manuscript fully available?**

Reviewer #1: Yes

Reviewer #2: Yes

PLOS authors have the option to publish the peer review history of their article (what does this mean? ). If published, this will include your full peer review and any attached files.

**Do you want your identity to be public for this peer review?** For information about this choice, including consent withdrawal, please see our Privacy Policy .

Reviewer #1: No

Reviewer #2: No

---

## [Editor Report · Acceptance letter]

PCOMPBIOL-D-25-00496R1

Task-relevant representational spaces in human memory traces

Dear Dr Heinen,

I am pleased to inform you that your manuscript has been formally accepted for publication in PLOS Computational Biology. Your manuscript is now with our production department and you will be notified of the publication date in due course.

With kind regards,

Zsofia Freund
